# *β*-Lactam Pharmacokinetic/Pharmacodynamic Target Attainment in Intensive Care Unit Patients: A Prospective, Observational, Cohort Study

**DOI:** 10.3390/antibiotics12081289

**Published:** 2023-08-05

**Authors:** Romain Guilhaumou, Constance Chevrier, Jean Loup Setti, Elisabeth Jouve, Amélie Marsot, Nathan Julian, Olivier Blin, Pierre Simeone, David Lagier, Djamel Mokart, Nicolas Bruder, Marc Garnier, Lionel Velly

**Affiliations:** 1Department of Clinical Pharmacology and Pharmacosurveillance, La Timone University Hospital; 13005 Marseille, France; 2Institut de Neurosciences des Systèmes, Aix Marseille University, INSERM UMR 1106, 13005 Marseille, France; 3University Hospital Timone, Department of Anaesthesiology and Critical Care Medicine, APHM, Aix Marseille University, 13005 Marseille, France; jeanloup.setti@ap-hm.fr (J.L.S.); pierre.simeone@ap-hm.fr (P.S.); david.lagier@ap-hm.fr (D.L.);; 4Faculté de Pharmacie, Université de Montréal, Montreal, QC H3T 1J4, Canada; amelie.marsot@umontreal.ca; 5Inst Neurosci Timone, INT, CNRS, Aix Marseille University, UMR7289, 13005 Marseille, France; 6C2VN, Inserm 1263, Inra 1260, Aix Marseille Université, 13005 Marseille, France; 7Department of Anaesthesiology and Critical Care Medicine, Institut Paoli-Calmette, 13009 Marseille, France; 8Sorbonne University, GRC29, APHP, DMU DREAM, Rive Droite, Site Tenon, 75020 Paris, France; 9Département d’Anesthésie-Réanimation et Médecine Périopératoire, CHU de Clermont-Ferrand, University Clermont Auvergne, 63000 Clermont-Ferrand, France

**Keywords:** dosing, antimicrobial, therapeutic drug monitoring, intensive care, pharmacodynamics, pharmacokinetics

## Abstract

Background: The aims of this study were to describe pharmacokinetic/pharmacodynamic target attainment in intensive care unit (ICU) patients treated with continuously infused *ß*-lactam antibiotics, their associated covariates, and the impact of dosage adjustment. Methods: This prospective, observational, cohort study was performed in three ICUs. Four *ß*-lactams were continuously infused, and therapeutic drug monitoring (TDM) was performed at days 1, 4, and 7. The primary pharmacokinetic/pharmacodynamic target was an unbound *ß*-lactam plasma concentration four times above the bacteria’s minimal inhibitory concentration during the whole dosing interval. The demographic and clinical covariates associated with target attainment were evaluated. Results: A total of 170 patients were included (426 blood samples). The percentages of empirical *ß*-lactam underdosing at D1 were 66% for cefepime, 43% for cefotaxime, 47% for ceftazidime, and 14% for meropenem. Indexed creatinine clearance was independently associated with treatment underdose if increased (adjusted odds ratio per unit, 1.01; 95% CI, 1.00 to 1.01; *p* = 0.014) or overdose if decreased (adjusted odds ratio per unit, 0.95; 95% CI, 0.94 to 0.97; *p* < 0.001). Pharmacokinetic/pharmacodynamic target attainment was significantly increased after *ß*-lactam dosage adjustment between day 1 and day 4 vs. no adjustment (53.1% vs. 26.2%; *p* = 0.018). Conclusions: This study increases our knowledge on the optimization of *ß*-lactam therapy in ICU patients. A large inter- and intra-patient variability in plasmatic concentrations was observed, leading to inadequate exposure. A combined indexed creatinine clearance and TDM approach enables adequate dosing for better pharmacokinetic/pharmacodynamic target attainment.

## 1. Introduction

Infection is a major source of morbidity and mortality in intensive care unit (ICU) patients. *ß*-lactam antibiotics are the most widely used antibiotic class in critical care worldwide; for instance, they account for 71% of antibiotics administered to ICU patients in France [1]. However, the severity of infections, the frequent antimicrobial resistance encountered in bacteria responsible for ICU-acquired infections, and the numerous pathophysiological changes causing specific pharmacokinetic (PK) and pharmacodynamic (PD) modifications make the use of *ß*-lactam antibiotic therapy in ICU patients particularly challenging [2]. Indeed, numerous studies have demonstrated that antibiotic plasma concentrations are often variable and unpredictable in this population [3,4,5]. Taking into account that every hour of delay in the administration of appropriate antibacterial therapy is associated with increased mortality in the most severe patients [6,7,8], determining the best empirical antimicrobial therapy regimen may be a real challenge in the ICU setting.

*ß*-lactam antibiotics are characterized by a time-dependent activity, defined as the time (T) the unbound (or plasma protein free, f) drug concentration remains above the minimal inhibitory concentration (MIC) during the dosing interval (%fT>MIC). In critically ill patients, a PK/PD target of %fT>4MIC = 100% has been recommended [9]. In order to achieve this goal, high daily doses of *ß*-lactam antibiotics are required. In addition, prolonged or continuous *ß*-lactam intravenous infusion seems to be the best administration modality to attain this PK/PD target, especially to treat microorganisms with a high MIC [10,11,12,13,14,15]. However, despite higher dosing regimens than those used in non-ICU patients, a significant number of ICU patients do not reach the plasmatic PK/PD target. Among the factors involved in inter-individual variability of *ß*-lactam concentrations, renal clearance appears to be the main contributor [16]. However, factors leading to intra-individual variability during the treatment course remain poorly described. Thus, the unpredictability of ICU patients’ exposure to *ß*-lactam antibiotics for a given dose (inter-individual variability) and over time (intra-individual variability) makes therapeutic drug monitoring (TDM) an important tool to avoid drug under- or overdosing [17,18,19,20].

The aims of this prospective, observational study were to determine (1) whether continuous *ß*-lactam infusion in critically ill patients allows PK/PD target concentrations to be reached; (2) the covariates associated with non-attainment of concentration targets; and (3) the impact of dosage adjustment on target attainment.

## 2. Results

### 2.1. Demographic and Clinical Data

During the study period, 170 patients receiving *ß*-lactam antibiotics were included, corresponding to 196 antibiotic treatments (26 patients presented two episodes of infection). A total of 426 blood samples were analyzed. The flow chart of the study is shown in Figure 1. The demographic characteristics of the 170 patients are described in Table 1. Bacteria were isolated in 136 (69.4%) out of the 196 suspected infections (Table 1). The most frequently isolated bacteria were *Escherichia coli* (26.1%), followed by *Pseudomonas aeruginosa* (17.0%) and *Klebsiella pneumoniae* (13.7%) (Appendix A).

### 2.2. Target Attainment at D1, D4, and D7

Among the 196 empirical *ß*-lactam treatments, 80 (40.8%) consisted of cefotaxime, 45 (23.0%) of cefepime, 41 (20.9%) of ceftazidime, and 30 (15.3%) of meropenem (Table 1). The concentrations of each antibiotic at D1, D4, and D7 are presented in Figure 2. The whole repartitions of under-, normo-, and overdosing at D1, D4, and D7 are presented in Table 2. Of note, the percentages of empirical *ß*-lactam underdosing at D1 were 66% for cefepime, 43% for cefotaxime, 47% for ceftazidime, and 14% for meropenem. The patients treated with cefotaxime, cefepime, and ceftazidime were more frequently underdosed than were those treated with meropenem (*p* < 0.01). Considering the 136 documented infections, *ß*-lactam concentrations were secondarily compared to the PK/PD target based on the specific ECOFF of the isolated bacteria. The percentages of *ß*-lactam underdosing at D1 were 39% for cefepime, 4% for cefotaxime, 48% for ceftazidime, and 16% for meropenem (Appendix A). 

### 2.3. Covariates Associated with Target Attainment

In the univariate analysis, serum Cr (CrS) and iCL_Cr_ were significantly associated with target attainment at D1 (Appendix A). At D4 and D7, iCL_Cr_ was associated with target attainment only for patients without dosage adjustment. In the multivariate analysis, iCL_Cr_ remained independently associated with target attainment (Appendix A). The risk of being overdosed decreased by 5% for each increase of 1 mL/min/1.73 m^2^ in iCL_Cr_ (adjusted odds ratio (aOR) per unit 0.95, 95% CI (0.94–0.97); *p* < 0.001). A threshold of iCL_Cr_ ≤ 43 mL/min/m^2^ predicted *ß*-lactam overdosing with a sensibility of 83% and specificity of 77% (ROC_AUC_ 0.85, 95% CI (0.80–0.88)) (Figure 3A). Conversely, the risk of being underdosed increase by 1% for each increase of 1 mL/min/1.73 m^2^ in iCL_Cr_ (aOR 1.01, 95% CI (1.00–1.01); *p* = 0.014). A threshold of iCL_Cr_ > 79 mL/min/m^2^ predicted *ß*-lactam underdosing with a sensibility of 56.9% and specificity of 74.4% (ROC_AUC_ 0.70, 95% CI (0.64–0.75)) (Figure 3B).

Concerning intra-patient variability over time, a change in CrS was the only factor associated with target attainment. A decrease in CrS between D1 and D4 was correlated with *ß*-lactam underdosing at D4 (best cut-off value to predict underdosing: −19 μmol/L, *p <* 0.001). Similar results were found between D4 and D7 (best cut-off value to predict underdosing: −3.0 μmol/L, *p* = 0.04).

### 2.4. Impact of Dosage Adjustment on Target Attainment

Thirty-eight dosage adjustments were observed between D1 and D4, and twenty-nine were observed between D4 and D7. The evolution of the target attainment between D1 and D4 and D4 and D7 for patients with and without dosage adjustment is represented in Figure 4. A dosage adjustment based on *ß*-lactam concentration measurement at D1 significantly increased the proportion of patients attaining the target at D4 (53% for patients with dosage adjustment vs. 26% for patients without, *p* = 0.02). By contrast, dosage adjustment based on *ß*-lactam concentration measurement at D4 did not modify the proportion of patients attaining the target at D7 (38% vs. 29%, *p* = 0.50). In the multivariate analysis, the occurrence of a dosage adjustment at D1 and/or D4 was associated with an increased target attainment at D4 and/or D7 (Appendix A).

### 2.5. Impact on Target Non-Attainment on Clinical Outcomes

In the multivariate analysis, no impact of target attainment at D1, D4, and D7 on SOFA score variation and duration of antibiotic treatment was observed. *ß*-lactam underdosing at D4 was associated with a trend toward an increased ICU length of stay compared with normodosing (37.5 (24–84.5) days vs. 33.0 (20.0–68.0) days, *p* = 0.051). 

## 3. Discussion

The main results of our study may be summarized as follows: (1) during the empirical phase of the antimicrobial therapy, 43%, 47%, 66%, and 14% of ICU patients treated with cefotaxime, cefepime, ceftazidime, and meropenem, respectively, were underdosed; (2) regarding the inter-individual variability, iCL_CR_ was the only parameter associated with target attainment; (3) regarding the intra-individual variability over time, the change in CrS was the only factor associated with target attainment; (4) TDM and *ß*-lactam dosage adjustment increased the proportion of ICU patients attaining the target, with a trend toward a decreased ICU length of stay.

Despite continuous administration of high-dose *ß*-lactam in the empirical phase, we observed that more than one-third of critically ill patients failed to attain the PK/PD target at D1. These results are consistent with those reported by Dhaese et al. [21]. In a cohort of 253 ICU patients, target attainment of 100%fT>4MIC during the first two days of antibiotic therapy was observed in only 37% and 75% of patients treated with piperacillin and meropenem, respectively. Similarly, Abdulla et al. reported only 37% of ICU patients attaining the 100%fT>4MIC target [22]. Interestingly, in our study, like in these two others, meropenem seemed to be the *ß*-lactam with the highest percentage of target attainment before any dosage adjustment. 

In the present study, approximately 70% of infections were documented and the PK/PD target was then secondarily adapted to the ECOFF of the isolated bacteria. Using this adapted PK/PD target did not decrease the proportion of ceftazidime and meropenem underdosing at D1 (48% vs. 47% and 16% vs. 14%, respectively), and conversely resulted in decreased cefepime and cefotaxime underdosage rates (39% vs. 66% and 4% vs. 43%, respectively). This latter difference is probably explained by the gap between the highest ECOFF value considered for the PK/PD target of the empirical phase and the ECOFF of the bacterial species that was actually isolated. Indeed, the “worst-case scenario” (i.e., an ECOFF of 4 mg/L for an infection due to *Staphylococcus aureus*) was observed in only 7 out of the 61 documented infections treated with cefotaxime, and similarly in only 5/44 documented infections treated with cefepime (ECOFF of 8 mg/L for *P. aeruginosa*). Moreover, using *P. aeruginosa*’s ECOFF as the highest ECOFF for cefepime would result in target plasma concentrations which may exceed the neurotoxicity threshold [23,24]. Consequently, the guidelines proposed a cefepime target plasma concentration based on the *Enterobacteriaceae* breakpoint (1 mg/L) in non-documented infections [9].

The risk factors of the wide inter-individual pharmacokinetic variability observed in ICU patients were previously evaluated and renal clearance appeared to be the main contributor [21,25]. This was confirmed in our cohort, and we report that iCL_CR_ was the only parameter associated with target attainment. Furthermore, both an increase and a decrease in iCL_CR_ were correlated with target attainment failure, exposing the patient to *ß*-lactam under- and overdosing, respectively. The best thresholds correlating with overexposure and underexposure were, respectively, iCL_CR_ values of ≤43 and >79 mL/min/1.73 m^2^. These results are consistent with those observed by Abdulla et al. [22] in a cohort of 147 patients and reinforce the recommendation to measure the glomerular filtration rate by calculating iCL_CR_ based on urinary and plasmatic creatinine values at the onset of *ß*-lactam treatment in ICU patients [9]. 

Regarding the intra-individual variability, CrS modification between D1 and D4 and D4 and D7 was a risk factor for target non-attainment. CrS variations may be due to changes in renal clearance but also to fluid loading and to ICU-acquired sarcopenia [26]. These results confirm the large intra-individual PK variability previously reported in ICU patients [27,28] and highlight the fact that a patient with adequate antibiotic exposure at D1 could be underdosed at D4 without any treatment modification. This reinforces the recommendation to measure CrS and estimate the glomerular filtration rate every time the clinical condition of the patient changes significantly [9].

Owing to the important PK inter- and intra-individual variabilities, the individualization of *ß*-lactam dosage is required in critically ill patients. TDM has been reported to increase the proportion of patients attaining the PK/PD target, which may increase treatment efficiency and decrease the selection of bacteria with the highest MICs within the inoculum [17,29,30,31,32,33,34,35]. The results of the present study support the previous results as the proportion of patients within the therapeutic ranges at D4 and D7 increased when the empirical *ß*-lactam dosage was adjusted based on TDM. Although our study was not designed to assess the impact of dosage adjustment on the clinical outcome, we observed a trend towards an increased ICU length of stay in underdosed patients, as previously reported by Abdulla et al. [22]. These results are consistent with the study of Al-Shaer et al. [32] and emphasize the importance of conducting further randomized studies designed to confirm the specific impact of TDM on the prognosis of ICU patients [36,37]. 

Our study has several limitations. First, several blood samples were missed during the study period, which may have decreased the power of the analyses performed regarding inter- and intra-individual variabilities. However, our study performed on 170 patients, involving 196 treatments and 426 blood samples, is one of the largest among all the studies focusing on β-lactam pharmacokinetics in an ICU population. Moreover, the statistical analyses took into account the presence of missing values, for instance, analyzing the factors influencing PK variability only in the population with all the concerned kinetic points available. Second, the actual MICs were not available to define the PK/PD target and the specific ECOFF of the isolated bacteria was used. Consequently, this PKPD target may have been overestimated in some patients. Finally, we conducted a β-lactam dosage adjustment in only about 50% of patients presenting under- or overdosing. This highlights one limitation with TDM: in order to affect the patient’s care, TDM should induce rapid dosage adjustment; otherwise, measuring β-lactam concentrations on its own is of little interest. Consequently, the percentage of dosage adjustment observed in our “real-life” study must be improved. We believe that this is possible, notably by protocolizing the dosage adjustment depending on TDM results and/or by systematically providing therapeutic adjustment advice at the time of TDM result reporting.

## 4. Patients and Methods

### 4.1. Study Design

This study was a prospective, multicenter, observational trial conducted in 3 ICUs of the Timone University Hospital and Paoli Calmette Institute, Marseille, France. The study was approved by the ethics committee “Comité de Protection des Personnes Sud-Est I” (Ref 2017-A01446-47). The period of inclusion was October 2015 to May 2017, corresponding to the introduction of the *ß*-lactam continuous infusion protocol in the 3 ICUs and a period of inclusion of 20 months. Due to the non-interventional design of the study, the ethics committee waived the need for an individual’s written consent according to French law [38]. Patients or next of kin were orally informed of the goal and design of the study.

Patients were eligible for enrolment if they were (1) between 18 and 80 years of age, (2) treated with continuous intravenous infusion of cefepime, cefotaxime, ceftazidime, or meropenem for presumed or confirmed infection while manifesting a systemic inflammatory response syndrome, and (3) expected to stay in the ICU for at least 7 days. Exclusion criteria were pregnancy and/or lactation, patient’s discharge from the ICU or death within the first 3 days after inclusion, and opposition to participation in the study. Patients’ comorbidities and renal clearance did not constitute exclusion criteria. 

Empirical antimicrobial therapy was initiated according to local protocols depending on the site of the suspected infection. *ß*-lactams were administered by continuous intravenous infusion and the dosage protocol was common to the three ICUs (Appendix A). Preparation (type of syringe, solvent, and stability) of the *ß*-lactams was also protocolized according to data available from the literature and the European database on drug stability [39].

The following demographic and clinical data were prospectively collected: age, sex, gender, height, weight, body surface area, body mass index, 24 h Simplified Acute Physiology Severity Score II (SAPS II), type of infection, identified bacteria, date and time of antibiotic initiation, *ß*-lactam antibiotic type and dosage, and date and time of treatment termination. The following clinical and biological data were collected at 24 h (D1), 4 days (D4), and 7 days (D7) after treatment initiation: modified Sequential Organ Failure Assessment (SOFA) score, serum albumin, hematocrit, urinary creatinine (CrU), serum creatinine (CrS), urinary output, measurement of indexed creatinine clearance (iCL_Cr_ = (1.73/body surface area) × 24 h CrU × 24 h urine volume/24 h CrS), treatment with continuous renal replacement therapy or extracorporeal life support.

### 4.2. Target Attainment and Therapeutic Drug Monitoring (TDM)

PK/PD target attainment was set as 100%fT>4MIC [9] and was determined during the empirical phase of the treatment (i.e., when the bacteriological documentation was not yet available). Thus, we used the highest European Committee on Antimicrobial Susceptibility Testing (EUCAST) ECOFF value among the bacteria usually involved in the suspected/confirmed infection (worst-case scenario) as the hypothetical MIC target. To limit the toxicity of *ß*-lactam antibiotics, a steady-state free plasma concentration equal to 8–10 times the MIC was proposed as the upper value of the target [40] (Appendix A). 

Venous blood samples were collected in heparinized tubes 24 h, and 4 and 7 days after treatment initiation. Assays were performed by the university-affiliated pharmacological laboratory (Department of Clinical Pharmacology and Pharmacosurveillance of Timone University Hospital, Marseille, France) using high-performance liquid chromatography coupled to ultraviolet detection method. Samples were transported within 5 h of sampling to the pharmacological laboratory and were then centrifuged. If necessary, plasma samples were stored at −20 °C until analysis and MES buffer (4-morpholineethanesulfonic acid, 1M) was added to plasma (1/10) for meropenem quantification. The assay protocol was adapted from the method of Verdier et al. [41] and validated according to EMA guidelines [42]. The limit of quantification was 0.5 mg/L for all molecules. No analytical interference was observed with a panel of drugs usually used in ICU at a concentration of 100 μg/mL, and in 10 plasma samples of ICU patients not treated with *ß*-lactams.

*ß*-lactam concentration results were available within 24 h of sampling, from Monday to Friday. In the case of target non-attainment, dosage adaptation was proposed. 

### 4.3. Statistical Analysis

Continuous variables are presented as means and 95% confidence (95% CI) intervals or standard deviations. Ordinal variables are presented as medians and inter-quartile ranges [IQR], and categorical variables are presented as numbers and percentages. 

The Kruskal–Wallis test was used for continuous variables with a non-Gaussian distribution to determine the covariates associated with target attainment. Multivariate analysis was performed using a logistic regression model such as the Generalized Estimating Equation (GEE) model with a step-by-step downward regression method. A Receiver Operating Characteristic (ROC) curve was drawn to identify the best cut-off value associated with target attainment. A mixed model was used to show the impact of intra-individual variability on target attainment. 

To assess the impact of TDM on target attainment, univariate analysis was performed using the chi-squared test or the Fisher exact test, when appropriate. A multivariate analysis using a GEE approach with the log function was performed, including the visit (D1, D4 and D7) and the existence or not of a dosage adjustment.

The impact of target non-attainment at D1, D4 or D7 on outcome criteria (SOFA score variation, duration of antibiotic treatment and length of stay in ICU) was also studied using a multivariate analysis using a GEE model and top-down process. The parameters included in the model were age, delay in the initiation of antibiotic treatment from admission, iCL_CR_ at D1, dosage adjustments, and target attainment. 

Statistical analysis was conducted using SAS software Version 9.4 (SAS Institute, Cary, NC, USA).

## 5. Conclusions

In conclusion, this study increased our knowledge on the optimization of *ß*-lactam therapy in ICU patients, with a focus on the importance of taking intra-individual variability into account. Beyond a high-dose initiation, *ß*-lactam therapy must be personalized and adjusted throughout the treatment due to the large inter- and intra-patient variabilities leading to relatively low proportions of PK/PD target attainment in ICU patients. A TDM-based approach of dosage adaptation, with regular iCL_Cr_ monitoring, is a method to contend with PK variability and improve antibiotic exposure. Further studies are required to definitively ascertain the impact of TDM with dosage adjustment on the prognosis of critically ill patients. 

## Figures and Tables

**Figure 1 antibiotics-12-01289-f001:**
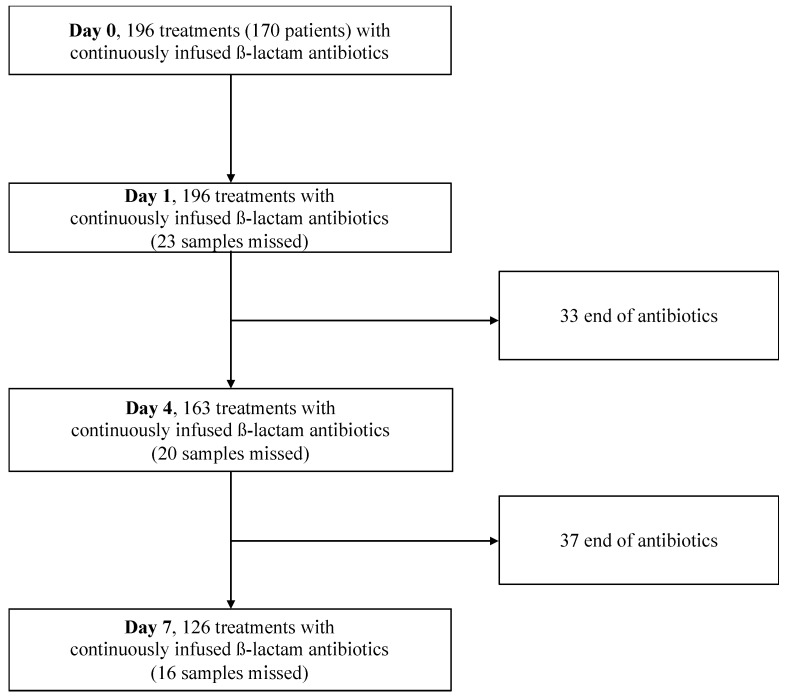
Study flow chart of 20-month observational study. Patients with concentration data at D1 and D4: n = 133; patients with concentration data at D4 and D7: n = 100; patients with concentration data at D1, D4, and D7: n = 95.

**Figure 2 antibiotics-12-01289-f002:**
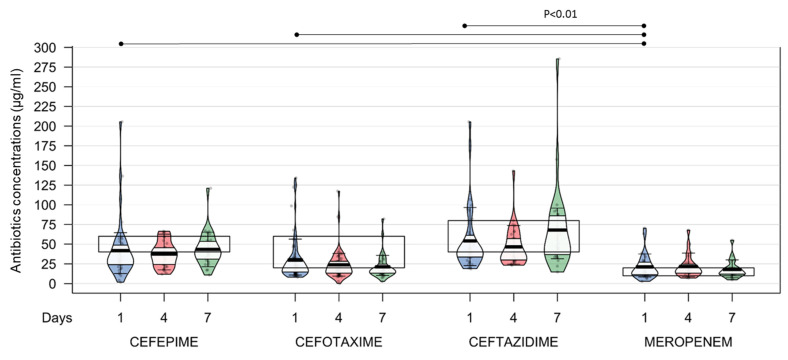
Observed concentrations of each antibiotic at D1, D4, and D7. Solid rectangles represent defined target concentrations.

**Figure 3 antibiotics-12-01289-f003:**
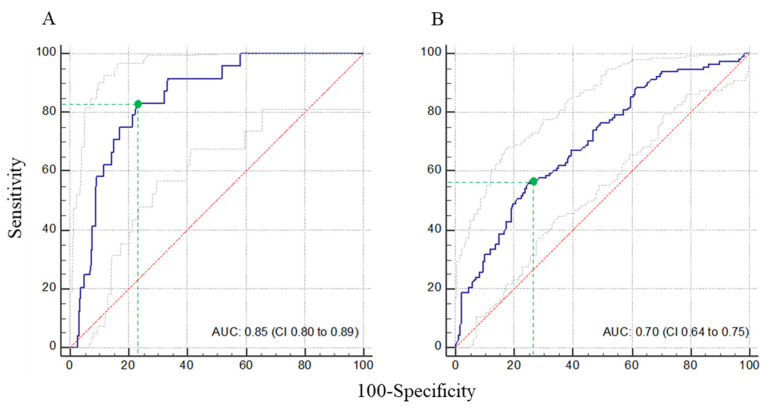
Receiver operating characteristic (ROC) curves of indexed creatinine clearance associated with overdosed (**A**) and underdosed (**B**) β-lactam antibiotics (blue lines). The optimal cutoff values for *ß*-lactam overdosing and underdosing calculated from these ROC curves were iCL_CR_ ≤ 43 mL/min/m^2^ (sensitivity 83.3% and specificity 77.3%) and iCL_CR_ ≥ 79 mL/min/m^2^ (sensitivity 56.9% and specificity 74.4%), respectively (green dots and dotted lines). Gray lines: 95% confidence interval (CI) bounds. Red line: identity line. AUC, area under curve.

**Figure 4 antibiotics-12-01289-f004:**
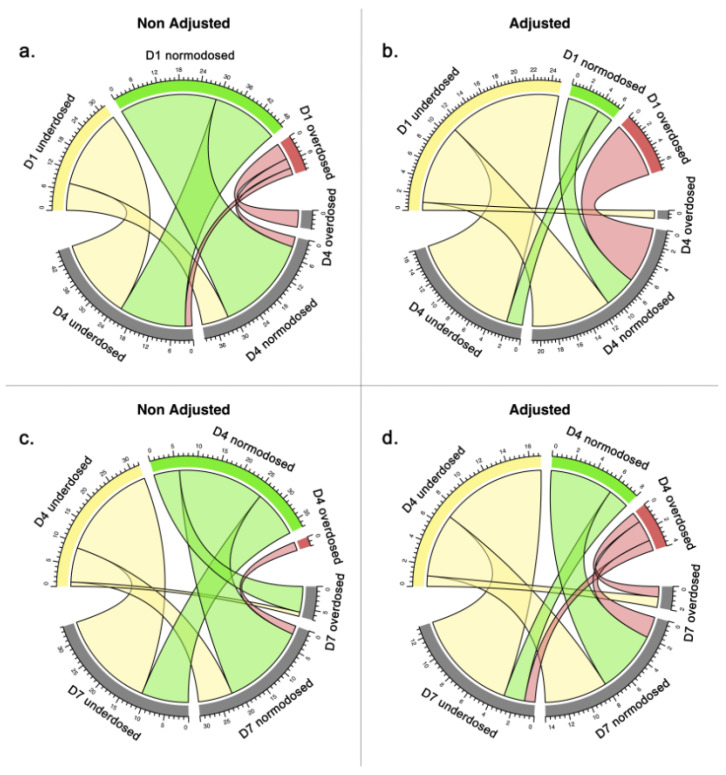
Evolution of patients’ target attainment between D1 and D4 (**a**,**b**) and D4 and D7 (**c**,**d**), with or without dosage adjustment (yellow: underdosed; green: normodosed, red: overdosed).

**Table 1 antibiotics-12-01289-t001:** Patient characteristics.

**Demographics (n = 170)**	
Age in years, median (IQR)	56.0 (53.6–58.4)
Male, n (%)	110 (64.7)
Weight in kg, mean ± SD	75 ± 19
Height in cm, median (IQR)	170 (165–178)
Body surface area in m^2^, median (IQR)	1.9 (1.7–2.0)
Body mass index in kg/m^2^, median (IQR)	25.4 (22.0–27.8)
Severity scores	
SAPS-2 score at ICU admission, median (IQR)	39 (38.5–51)
Modified SOFA score at day 1, median (IQR)	4 (2–5)
Renal replacement therapy, n (%)	12 (6)
Extra corporeal life support, n (%)	6 (3)
**Biological data at day 0 (n = 196), median (IQR)**	
Hematocrit in %	29 (26–34)
Serum albumin in g/L	29 (25–33)
Serum creatinine in μmol/L	65 (48–104)
Indexed creatinine clearance in mL/min/1.73 m^2^	71.4 (31.5–111.0)
**Infection (n = 196)**	
Documented, n (%)	136 (69.4)
Respiratory tract	70 (51.5)
Primary bacteremia	35 (25.7)
Urinary tract	21 (15.5)
Intra-abdominal	1 (0.7)
Central nervous system	3 (2.2)
Other sites	6 (4.4)
Not documented, n (%)	60 (30.6)
**Antibiotic treatments (n = 196)**	
Empirical treatment, n (%)	
Cefotaxime	80 (40.8)
Cefepime	45 (23.0)
Ceftazidime	41 (20.9)
Meropenem	30 (15.3)
Duration of antibiotic therapy in days, median (IQR)	7 (5–10)
**Outcome (n = 170)**	
Length of ICU stay in days, median (IQR)	33 (17–67)
Alive at day 28, n (%)	143 (84.1)

Data are number (%) or median (inter-quartile range). SAPS-2, Simplified Acute Physiology Score 2; SOFA, Sepsis-related Organ Failure Assessment.

**Table 2 antibiotics-12-01289-t002:** Target attainment at D1, D4, and D7.

		Patient Target Attainment (%)					
		Day 1	n	Day 4	n	Day 7	n
Cefepime	underdosed	66	38	62	34	52	27
	normodosed	24		38		44	
	overdosed	10		-		4	
Cefotaxime	underdosed	43	69	51	56	71	36
	normodosed	51		45		26	
	overdosed	6		4		3	
Ceftazidime	underdosed	47	37	48	30	30	27
	normodosed	42		48		59	
	overdosed	11		4		11	
Meropenem	underdosed	14	29	17	23	20	20
	normodosed	57		57		60	
	overdosed	29		26		20	

## Data Availability

The datasets used and/or analyzed during the current study are available from the corresponding author on reasonable request.

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
