# Peer review of "β*-Lactam Pharmacokinetic/Pharmacodynamic Target Attainment in Intensive Care Unit Patients: A Prospective, Observational, Cohort Study"

_antibiotics, 2023, doi:10.3390/antibiotics12081289_

Round 1

Reviewer 1 Report

The authors submitted manuscript in the title of, ß-Lactam pharmacokinetic/pharmacodynamic target attainment in intensive care unit patients: a prospective observational cohort study. The authors should consider the followings:

In Table Patients characteristics, please explain why the number of demographics (n=170) was lower than that of infection (n=196).

Please explain the results of, aOR 1.01 or 0.95, which are very close to 1.0.

The authors should provide the rationales and workflow of sample size estimation in this observation study. 

Please define and set number to the tables and figures, in the manuscript.

The authors should use a better form of data presentation, instead of in Figure Evolution of patients target attainment between D1-D4 (a, b) and D4-D7 (c, d), owing to dosage adjustment or not.

Please revise that, in Y axis should be sensitivity, In Figure Receiver operating characteristic (ROC) curves of indexed creatinine clearance associated with β-lactam antibitiotics overdosed (A) and underdosed (B).

In Figure "Observed concentrations of each antibiotic at D1, D4 and DSolid rectangles represent defined target concentrations", the authors should denote in the figure any statistically significant results.

Please provide further elaboration on the study limitations in this study.

The authors should state the novelty of findings in this study, in abstract and conclusion part.

Did the authors document the oral informed processes of, "Patients or next of kin were orally informed of the goal and design of the study".

The authors may elaborate on how they design the length of study period, for why in particular, October 2015 to May 2017. 

The authors should find english language professional to proofread carefully for this manuscript.

Since the study used hplc/uv for detection, did the authors validate the assay for high bilirubin state, and/or other interference studies.

Extensive editing of English language required

Author Response

Answer to reviewer’s comments

Title of the manuscript: “ß-Lactam pharmacokinetic/pharmacodynamic target attainment in intensive care unit patients: a prospective, observational, cohort study."

We thank the reviewers for their comments. Please find below our point-by-point answers to these comments. Modifications in the revised manuscript are highlighted in yellow.

Reviewer 1.

The authors submitted manuscript in the title of, ß-Lactam pharmacokinetic/pharmacodynamic target attainment in intensive care unit patients: a prospective observational cohort study. The authors should consider the followings:

In Table Patients characteristics, please explain why the number of demographics (n=170) was lower than that of infection (n=196).

26 patients presented two episodes of infection. This point is described in the Results-Demographic and clinical data section.

Please explain the results of, aOR 1.01 or 0.95, which are very close to 1.0.

These results can be explained by the fact that we have presented results using adjusted odds ratio (aOR) per unit (continuous variable) and not "conventional" OR (from categorial). Results are then weighted for an increase/decrease of 1 mL/min/1.73m2 in iCLCr..

The authors should provide the rationales and workflow of sample size estimation in this observation study.

We agree with the referee that a sample size estimation would be the most appropriate methodology. However, at the time of the beginning of the study, there were few data on the PKPD targets of β-lactam antibiotics in intensive care patients, using continuous infusion. Consequently, it was difficult for us to formulate hypotheses on the proportions of patients in the targets at D1, D4 and D7 and determine the sample size estimation. The period of inclusion was estimated at 20 months and at the end of the inclusion period, a sample size estimation was secondarily calculated using observed results. The hypothesis were: an alpha risk of 5% and an expected power of 80% in a bilateral situation; expected PKPD target attainment of 50%, a correlation matrix between repeated data of autoregressive order 1, a drop-out rate of 30% between D1 and D4, and 50% between D1 and D7. The number of patient was then estimated to 120 to observe the impact of covariates and TDM on PKPD target attainment.

Please define and set number to the tables and figures, in the manuscript.

We thank the reviewer for this point and number of tables and figures have been added in the whole manuscript.

The authors should use a better form of data presentation, instead of in Figure Evolution of patients target attainment between D1-D4 (a, b) and D4-D7 (c, d), owing to dosage adjustment or not.

We thank the reviewer for this proposition. However, we have chosen to present the results in this way, in order to describe intra-individual PK variability, i.e. the evolution of PKPD target attainment for a same patient over the course and the impact of dose adjustment.

Please revise that, in Y axis should be sensitivity, In Figure Receiver operating characteristic (ROC) curves of indexed creatinine clearance associated with β-lactam antibitiotics overdosed (A) and underdosed (B).

Figure 3 has been corrected.

In Figure "Observed concentrations of each antibiotic at D1, D4 and D7. Solid rectangles represent defined target concentrations", the authors should denote in the figure any statistically significant results.

We agree with the referee that denoted in the figure any statistically significant results is an important point. Figure 2 has been modify.

Please provide further elaboration on the study limitations in this study.

We thank the referees for this point and study limitations have been developed in the discussion section.

The authors should state the novelty of findings in this study, in abstract and conclusion part.

Novelty of findings in the study have been specify in the abstract and the conclusion.

Did the authors document the oral informed processes of, "Patients or next of kin were orally informed of the goal and design of the study".

Information on the study was provided to patients or next of kin at the time of the start of the treatment with β-lactam. However, this process was not documented.

The authors may elaborate on how they design the length of study period, for why in particular, October 2015 to May 2017. 

The period of inclusion was October 2015 to May 2017, corresponding to the introduction of the ß-lactam continuous infusion protocol in the 3 ICUs and a period of inclusion of 20 months. This point has been clarify in the manuscript.

The authors should find english language professional to proofread carefully for this manuscript.

Manuscript has been reviewed by MDPI editing service.

Since the study used hplc/uv for detection, did the authors validate the assay for high bilirubin state, and/or other interference studies.

We thank the referee for this point and information of interference studies during the process of the validation of the analytical method have been added in the manuscript.

Reviewer 2 Report

From the beginning, I want to congratulate the authors for the topic addressed. This is of major interest, at a time when antibiotics are heavily administered, especially in critically ill patients.

Considering that I am first and foremost a clinician, I will mainly insist on clinical data, although the approach is related to pharmacokinetics and pharmacodynamics.

Critically ill patients have multiple causes of decompensation, some receiving parenteral antibiotics only prophylactically, with no clear evidence of infection. I would like to know if they were also included in the study

It was not stated how many of the critically ill patients had sepsis and whether their outcome differed from patients who did not have sepsis related to reaching the target dose.

I would like to know if the possible correlations between PK/PD target attainment and the degree of sensitivity of the pathogen to the administered beta-lactam have been studied.

Author Response

Answer to reviewer’s comments

Title of the manuscript: “ß-Lactam pharmacokinetic/pharmacodynamic target attainment in intensive care unit patients: a prospective, observational, cohort study."

We thank the reviewers for their comments. Please find below our point-by-point answers to these comments. Modifications in the revised manuscript are highlighted in yellow.

Reviewer 2.

From the beginning, I want to congratulate the authors for the topic addressed. This is of major interest, at a time when antibiotics are heavily administered, especially in critically ill patients.

Considering that I am first and foremost a clinician, I will mainly insist on clinical data, although the approach is related to pharmacokinetics and pharmacodynamics.

Critically ill patients have multiple causes of decompensation, some receiving parenteral antibiotics only prophylactically, with no clear evidence of infection. I would like to know if they were also included in the study. We thank the referee for this remark. As describes in the study design section, only patients treated for an infection were included. Thus, patient treated only prophylactically were not included.

It was not stated how many of the critically ill patients had sepsis and whether their outcome differed from patients who did not have sepsis related to reaching the target dose. Thank you for your feedback. All patients included in the study had sepsis. This point was clarified in the inclusion criteria by adding the notion of "treated for presumed or confirmed infection while manifesting a systemic inflammatory response syndrome", corresponding to the definition of sepsis in effect at the time of the inclusion of patients in this study.

I would like to know if the possible correlations between PK/PD target attainment and the degree of sensitivity of the pathogen to the administered beta-lactam have been studied. Correlations between PK/PD target attainment and the degree of sensitivity of the pathogen have been studied for documented infections. Indeed, ß-lactam concentrations were secondarily compared to the PK/PD target based on the specific ECOFF of the isolated bacteria. Results are described in the Supplementary Table S4 and discuss in the discussion section.

" In the present study, approximately 70% of infections were documented and the PK/PD target was then secondarily adapted to the ECOFF of the isolated bacteria. Using this adapted PK/PD target did not decrease the proportion of ceftazidime and meropenem underdosing at D1 (48% vs. 47% and 16% vs. 14%, respectively), and conversely resulted in decreased cefepime and cefotaxime underdosage rates (39% vs. 66% and 4% vs. 43%, respectively). This latter difference is probably explained by the gap between the highest ECOFF value considered for the PK/PD target of the empirical phase and the ECOFF of the bacterial species that was actually isolated. Indeed, the “worst-case scenario” (i.e., an ECOFF of 4 mg/L for an infection due to S. aureus) was observed in only 7 out of the 61 documented infections treated with cefotaxime, and similarly in only 5/44 documented infections treated with cefepime (ECOFF of 8 mg/L for P. aeruginosa). Moreover, using P. aeruginosa’s ECOFF as the highest ECOFF for cefepime would result in target plasma concentrations which may exceed the neurotoxicity threshold [28,29]. Consequently, the guidelines proposed a cefepime target plasma concentration based on the Enterobacteriaceae breakpoint (1 mg/L) in non-documented infections [9]".
